# Evacuation Time Estimation Model in Large Buildings Based on Individual Characteristics and Real-Time Congestion Situation of Evacuation Exit

Qing Deng [1], Bo Zhang [2], Zheng Zhou [3], Hongyu Deng [3], Liang Zhou [3], Zhengqing Zhou [3] and Huiling Jiang [4,*]

1   Research Institute of Macro-Safety Science, University of Science and Technology Beijing, Beijing 100083, China
2   Xi'an Fire Rescue Detachment, Xi'an 710000, China
3   School of Civil and Resource Engineering, University of Science and Technology Beijing, Beijing 100083, China
4   Technical Support Center for Prevention and Control of Disastrous Accidents in Metal Smelting, University of Science and Technology Beijing, Beijing 100083, China
*   Correspondence: jianghuiling@ustb.edu.cn

**Abstract:** Fire is one of the most common and harmful disasters in real life. In 2021, firefighting teams in China reported 748,000 fires, resulting in 1987 deaths, 2225 injuries and CNY 6.75 billion of direct property losses, which account for 0.05‰ of GDP. Scientific and accurate estimation of evacuation time can provide decision support for intelligent fire evacuation. This paper aims to effectively improve the evacuation efficiency of people in large buildings, especially for a scenario with intricate evacuation passages. There are many factors that make a difference in evacuation time, such as individual behavior, occupant density, exit width, and so on. The people distribution density is introduced to effectively assess the impact of unstable pedestrian flow and unbalanced distribution in the process of evacuation. The verification results show that there is a strong positive correlation between people distribution density and evacuation time. Combining the people distribution density with many other factors, the training dataset is built by Pathfinder to learn the relationship between evacuation time and influencing factors. Finally, an evacuation time prediction model is established to estimate the consumption time that occupants spend on moving in the evacuation process based on stacking integration. The model can assist occupants in choosing different channels for evacuation in advance. After testing, the average error between the predicted evacuation consumption time and the reference time is 3.63 s. The result illustrates that the model can accurately predict the time consumed in the process of evacuation.

**Keywords:** evacuation time estimation; buildings fire; people distribution density; Pathfinder; stacking integration

## 1. Introduction

Fire is one of the most common and harmful disasters in real life and is directly related to people's lives and property. In 2021, firefighting teams in China reported 748,000 fires, resulting in 1987 deaths, 2225 injuries and CNY 6.75 billion of direct property losses, which account for 0.05‰ of GDP. Minimizing the evacuation time is one of the most effective measures to reducing the final casualty consequences [1]. Many real fire disasters have illustrated the importance of minimizing evacuation time. For example, 120 people were burned because of complex and rare escape routes in the fires of the Jilin poultry company on 3 June 2013 [2]. Six fatalities and a dozen injuries in the corridor were choked by smoke in the Cook County, Chicago, Administration Building fire on 17 October 2003 [3]. Therefore, how to improve evacuation efficiency based on the accurate estimation of evacuation time has become a prerequisite for safe evacuation.

The study of evacuation time has attracted much attention. The occupants inside the fire site only have a short amount of time to evacuate, which is called the golden evacuation

time [4–6]. When the available evacuation time ($T_{ASE}$) is longer than the required evacuation time ($T_{RSE}$), occupants can evacuate to safe areas, as shown in Figure 1. It has been proven that total required evacuation time consists of awareness time, pre-movement time, and evacuation time [4,7–10]. Evacuation time is the sum of movement time and waiting time during the evacuation process.

$$T_{EVA} = T_{MOV} + T_{WAIT}$$

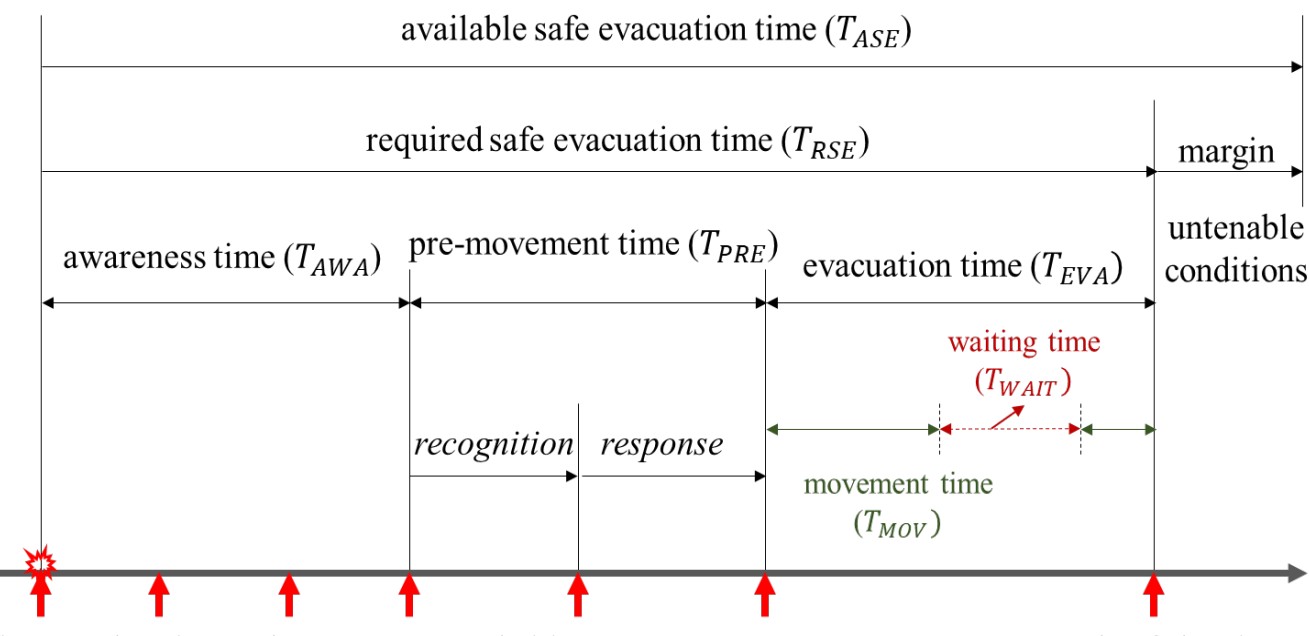

**Figure 1.** The composition of evacuation time: fire start: ignition (the point when a fire starts); detection: the point when the detection systems are activated; i.e., sprinklers; alarm: the point when the alarm is sounded; awareness: the point when occupants recognize that an emergency situation is taking place; decision to act: the point when occupants recognize that the emergency situation is a fire accident; start to move: the point when occupants respond to the situation and begin escape movement; reach safe location: the point when occupants reach safe location; untenable conditions: the point when fire components, such as smoke, heat, toxic gases, narcotic gases, and irritant gases, kill the occupants. The pre-movement time is also known as pre-evacuation time.

Each part of the evacuation time has its unique characteristics, which is the base of different estimation methods. A widely used method is to build some estimation models based on the identification of various factors. Therefore, factor identification is the first step for estimation model building. There are different factors to consider and different methods to use at each stage of evacuation, as shown in Table 1.

**Table 1.** The summary of evacuation time at different stages.

| References | Stages | Explanation of Each Stage | Factors | Methods |
|---|---|---|---|---|
| Ming-Kuan TSAI. (2015) [11], Grajdura, S (2021) [12], Kodur, VKR (2020) [13] | Awareness time | the time from the fire starts to the point when occupants recognize that an emergency is taking place | the efficiency of fire monitoring systems | image recognition and knowledge models |
| Kuligowski E. (2013) [14], Chu, GQ (2006) [7,15], Rogsch, C (2014) [16], Forssberg, M (2019) [17] | Pre-movement time | the time from the point when occupants recognize that an emergency is taking place to the point when the occupants start to move | individual behavior | the protective action decision model |
| W.K. Chow. (2007) [9], Fang, ZX (2011) [18] | Waiting time | the total time of occupants waiting in the evacuation process, similar to the movement behavior, waiting time is the total time spent waiting for multiple times. | occupant loadings | numerical simulations with BuildingEXODUS |
| YAN W D. (2021) [19], Kirik, E (2014) [20] Koo, J et al. (2012) [21] | Movement time | the time from the point when the occupants start to move to the point when occupants end the escape movement | gender, number, age, disability | Pathfinder software simulation |
| Lin C S et al. (2018) [4], Aleksandrov, M (2015) [22], Gao, H (2020) [23] | Evacuation time | the total time of occupants moving in the evacuation process; sometimes the evacuation process is intermittent; then, evacuation time is the total time of multiple movements | floor area, number of exits, and per-floor occupant load | a rapid prediction model based on the traditional Togawa model |
| Jiang Y L et al. (2021) [24], Xiao, MF (2022) [25], Gwynne, S (2012) [26], Chang-Jun (2021) [27] | Required safe evacuation time | the time from when the fire starts to the point when occupants reach a safe location | building structure and personnel distribution | multi-factor combined method |
| Tian F et al. (2019) [28], Tosolini, E (2012) [29] | Available safe evacuation time | the time from when the fire starts to the point when fire components, such as smoke, heat, toxic gases, narcotic gases, and irritant gases, kill the occupants | concentration of toxic gas, smoke layer height, and temperature, radiant heat flux | multi-factor combined method |

The influencing factors of evacuation time have been explored by many researchers. Different evacuation stages have different main factors. The awareness time is studied by building an intelligent construction site fire management platform [11]. The impact of individual behavior is analyzed on the pre-movement time using a protective action decision model [14]. GridFlow evacuation model and probability distribution are utilized to study the effect of pre-evacuation time on evacuation time prediction under different occupant densities and exit widths [7,15]. Chow, WK [9] studied the importance of waiting time in evacuation time prediction under crowded conditions. Pathfinder is a widely used software simulation to study movement time [19]. Lei et al. [30] explored the influence of rules about occupant density and exit width on evacuation efficiency. Each period of the above study is summarized to form the evacuation time. Furthermore, many factors have direct impacts on evacuation time prediction such as evacuation numbers, minimal group effect, pre-action time and the stability region [31,32]. Lin et al. [4] analyzed the impact of floor area, number of exits, and per-floor occupant load on evacuation time. The length, width, height and the number of exits are also factors influencing the evacuation efficiency in a building. Moreover, the practicability and correctness of the evacuation model need to be discussed by comparing the required safe evacuation time with the available safe evacuation time [24–29]. Fifteen pre-evacuation influencing factors have been

analyzed using Interpretive Structural Modeling, but all of them are static [8]. According to previous studies, evacuation time can be influenced by many factors, such as age, sex, speed, shoulder width, and others [33–35]. Age is utilized to estimate the moving speed and corresponding shoulder width [36]. Without considering the influence of fire and other factors, this paper considers two categories of factors, including individual characteristics and the dynamic congestion degree of the evacuation route. The better an occupant's escape ability and the evacuation route's environment, the less required evacuation time they need.

Different models or methods are applied to estimate evacuation time based on different influencing factors. Three popular methods are simulation models, machine learning approaches, and artificial neural network. Modeling and simulation tools are widely used in many different scenarios. AnyLogic is used to study emergency evacuation with unbalanced utilization of exits at the platform level [37]. A model was built with Pathfinder to combine the personnel escape behavior on mine fire [38]. Moreover, simulation modeling methods are often developed based on some classic models [11,12]. For example, numerical simulations with BuildingEXODUS are used to calculate the waiting time under two different densities of occupants [9]. The combination of numerical simulations with Pathfinder is also used to verify the impact of occupant number and sex on the movement time [4]. The powerful machine learning approaches can help scenario building for planning future facilities to provide a reference for evacuation time prediction. Machine learning approaches such as cellular automata model, multi-agent modeling, genetic algorithm (GA) with neural networks (NNs), and fuzzy logic (FL) are widely used to study pre-evacuation stage, earthquake casualty prediction, evacuation traffic prediction and real-time evacuation [39–44]. In addition, artificial neural network (ANN) is often used to study pre-evacuation behavior and available safe egress time prediction for its powerful data learning ability [44,45]. A deep neural network surrogate model is proposed to plan optimal evacuation routes to reduce casualties in toxic gas leak incidents [46]. The evacuation design simulation is implemented for subway station building based on the deep neural network model [47]. The methods mentioned above construct various models mainly based on many factors or parameters to predict evacuation time under various circumstances. However, most of them mainly focus on various static parameters. Few research studies consider dynamic changes during the evacuation process, especially the influence of unstable pedestrian flow and unbalanced distribution.

This study aims to build an evacuation time estimation model (ETEM) based on both static and dynamic factors in large buildings with intricate evacuation channels. Although evacuation time can be divided into many parts, this paper mainly focuses on the stages of movement and waiting during the evacuation process. The movement time can be determined by the occupants' moving ability and the different degrees of crowdedness in the crowded environment. The waiting time is determined by the movement speed of the crowd in front of you and the degree of congestion. Pedestrians are not uniformly distributed in the evacuation space, and their movement is also random and dynamic. The people distribution density is introduced to describe the impact of such unstable flow and uneven distribution on people's evacuation process. It is defined by some typical scenarios with unstable flow and an uneven distribution of occupants. The accuracy of the evacuation time estimation can be improved through the introduction of people distribution density in the prediction model. Various factors are analyzed by multi-methods, including some static factors and dynamic information. Combining these factors with the people distribution density, an evacuation prediction model is proposed to calculate the evacuation time through the stacking integration strategy. Finally, the effectiveness of our model can be verified with a small difference from the simulation time by Pathfinder.

## 2. Materials and Methods

### 2.1. Framework

The ETEM can estimate evacuation time in different scenarios by learning the correlation between the evacuation time and influencing factors based on the stacking integration method. The framework of ETEM consists of four phases, as shown in Figure 2. First, many factors influencing evacuation time are analyzed with a literature research and case analysis. These factors include individual characteristics of evacuees, evacuation environment, and the dynamic information. Second, datasets should be built to lay the foundation for correlation learning and model training and validation. Multi-scenarios are set to establish simulation models by Pathfinder. Then, the data corresponding to influencing factors and their corresponding evacuation time can be recorded to create the dataset. Within the dataset, 85% of them are randomly selected as the training data, and the remaining 15% are the test data. Third, the stacking integration method is applied to learn the relationship between evacuation time and factors on the training set. XGB, LGB, and GBoost classifiers are integrated into the stacking strategy to make the evacuation time prediction model. Finally, the test dataset is used to verify the accuracy of the model through the comparison with the real simulation results.

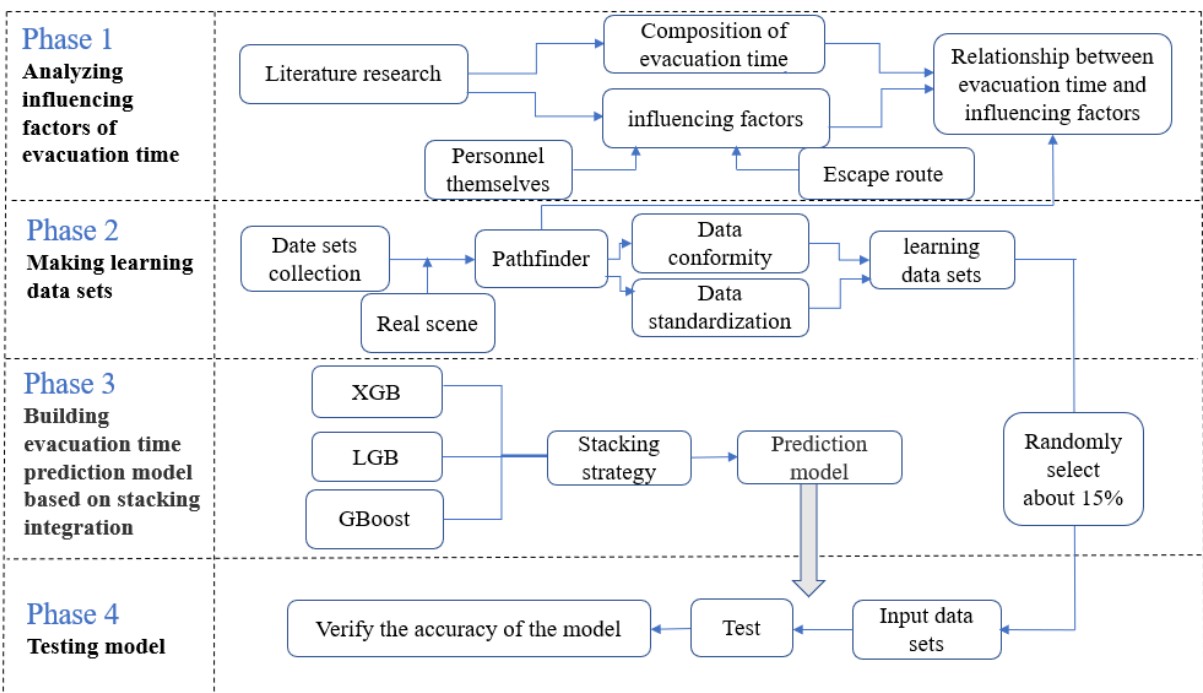

**Figure 2.** Development flow chart of evacuation time prediction model.

### 2.2. Stacking Integrating Method

Stacking integrating strategy is an integrated learning technology, which was proposed by Wolpert in 1992 [48]. The output of the first-layer classifiers is the input of the second-layer classifiers. New weak classifiers can be trained based on the negative gradient information of loss function. Then, the trained weak classifiers are combined into the existing model. Therefore, the generalization error of the learning process can be greatly reduced to show better nonlinear expression ability. Essentially, the goal of stacking is to weigh variance and reduce bias. In recent years, the stacking ensemble learning method has been applied to prediction problems in many fields with strong predictive ability. For example, the gradient lifting spatial model and neural network mechanism as the basic classifiers are proposed to effectively improve the prediction effect of pollutant concentration [49]. A stacking algorithm with multiple metamodels is applied to predict house price [50]. In the field of healthcare, stacking algorithms are also used for medical data integration in disease prediction [51,52].

In this paper, the stacking strategy is introduced to improve the accuracy of the prediction model through integrating three classifiers of XGB (extreme gradient boosting), LGB (light gradient boosting) and GBoost (gradient boosting). XGB is a supervised classification and regression algorithm. XGB is faster than other traditional gradient boosting algorithms. LGB has advantages of fast training efficiency, low memory usage, high precision, parallel learning support and strong ability to handle big data. To solve the problem of general loss function optimization, Freidman proposed a gradient boosting algorithm. The integration of these three classifiers can make all their advantages manifest and reduce their respective disadvantages.

### 2.3. People Distribution Density

People move in the evacuation route, and the pedestrian distribution is unbalanced and dynamic during the evacuation process. The optimum evacuation route cannot only be determined by the occupant distribution and location at the current time because occupants can change their movement behavior and direction at any time, resulting in a change of distribution density. The situation may happen that the route is not crowded when a person is going to move there, but the congestion may be caused by groups of occupants from other routes when they evacuate to there. A situation such as this can have a big influence on evacuation efficiency.

### 2.3.1. Scenario Analysis

The behaviors of pedestrians in the evacuation have some characteristics to influence path selection and evacuation efficiency in walking facilities. First, they will prefer to choose a path to the exit with a straight line for the shortest distance and minimum evacuation time. Second, they tend to flexibly change their evacuation path according to the surrounding environment. However, it is difficult to realize, especially when the situation is very crowded. Third, once there are obstacles in the evacuation space, most of the pedestrians choose to pass through them at a lower speed rather than bypassing them through other paths, which will influence evacuation efficiency. Therefore, the real evacuation scenario is very complex. All these dynamic human and environmental factors have big effects on evacuation time and efficiency. To be specific, environmental factors affect the people distribution density by influencing people's thoughts and behaviors. People's thoughts and behaviors are the direct and main causes, while environmental factors are the indirect and secondary causes. People distribution density is introduced to illustrate the coupling effect of these dynamic behavior. Two typical scenarios are used to enhance the understanding and to introduce the estimation of this concept.

In the evacuation situation, evacuation routes are not always straight and are often intricate. One of main reasons that evacuees do not choose effective routes is that they cannot see the flow of people on other intertwined routes. Figure 3 gives a simple example of two evacuation paths that are perpendicular to each other. The evacuees are marked by pink, and safety exits are marked by green rectangles. The evacuee (p) is closer to the left exit (A) from the straight-line distance, and the number of evacuees on the left side is much less than that on the right side. The evacuee (p) should select the left exit when the evacuee considers only the distance from the exit to the evacuees' distribution. However, there may be many evacuees moving from the vertical routes, causing congestion of the left side, which will greatly reduce the evacuation efficiency of the evacuee (p). Therefore, the selection of evacuation routes or exits cannot just consider the distance from the exits and the evacuee distribution in the current time. The dynamic pedestrian flow should also be considered in the evacuation.

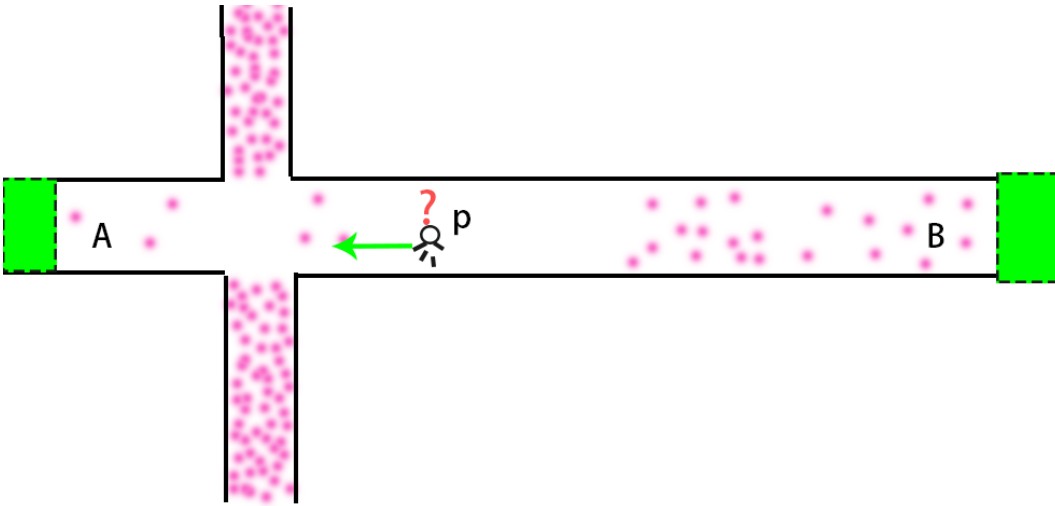

**Figure 3.** The influence of the dynamic flow of evacuees during evacuation.

The second typical scenario is a straight evacuation route, as shown in Figure 4. Suppose the degree of crowder in area A is the same as that in area B. Compared to the right side, the distance from the evacuee (p) to the left exit is longer, but the distance from the crowded zone to the left exit is much shorter. The right side may be the better choice when just considering the distance with the exit, the people density distribution in the route, and the distance with the crowded zone. However, the slow speed of the evacuees in the crowded zone with high density moving to the exit is a serious constraint during the actual evacuation.

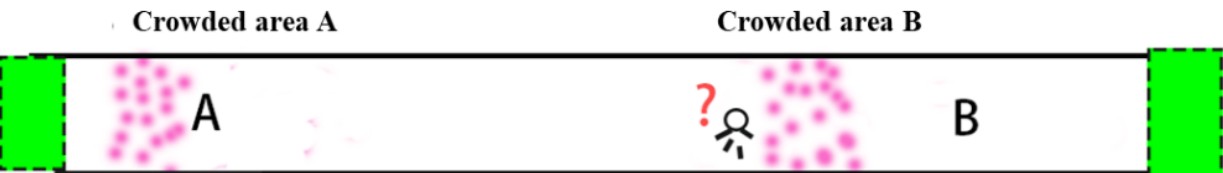

**Figure 4.** The unbalanced crowded zone.

2.3.2. Definition of People Distribution Density

Considering the previous two scenarios together, the concept of people distribution density is introduced to assess the comprehensive effect of the dynamic human and environmental factors on the evacuation process. Based on the above analysis, people distribution density is affected by the crowder degree, moving velocity of evacuees, and the distance to exits, except for the width of the evacuation route. For an independent area, the people distribution density (*P_dis_den*) in this area is defined by Equation (1).

$$P\_dis\_den = \sum_i \sum_j \frac{N_i}{V} \cdot \frac{L_{ij}}{\alpha \cdot d_{ij}} \tag{1}$$

where $V$ is the average velocity of occupants, and $N_i$ is the total number of occupants in the $i_{th}$ evacuation route. For an evacuation route, it can be divided into $j$ segments according to different widths because different widths mean that occupants need to turn when they move through the joints of different widths. $L_{ij}$ is the length of the $j_{th}$ segment for the $i_{th}$ evacuation route, and $d_{ij}$ is the width of the $j_{th}$ segment for the $i_{th}$ evacuation route.

In the actual evacuation, many occupants need to turn when they want to reach the emergency exit, i.e., the evacuees in the vertical routes in Figure 3. In the process of their turn, they tend to walk on one side, which results in a width loss of the evacuation channel. For example, as shown in Figure 3, the evacuees in the vertical routes will walk on the left side, rather than on the right side if they select the left exit to evacuate. $\alpha$ is

the effective coefficient of width. Very little research has been conducted to analyze the effective coefficient. Therefore, in order to obtain the more accurate coefficient, many simulation experiments have been conducted to determine the value of $\alpha$. These simulation experiments are compared by setting different values of $\alpha$ with an interval of 0.05 in the range of [0.50, 1.00]. The actual evacuation time is set as the assessment criterion. According to the comparison of simulation experiment results, only when the coefficient is set at 3/4 (0.75) are the simulation results consistent with the actual situation. Therefore, the effective width after a rectangular turn is set as 3/4 of the channel width in this study.

$$\alpha = \begin{cases} \frac{3}{4}, \text{the evacuation channel is not straight; occupants need to turn;} \\ 1, \text{the evacuation channel is straight; occupants do not need to turn.} \end{cases}$$

People density distribution does not always affect every evacuee during the evacuation process. It will play a role when some conditions are satisfied. Evacuation priority is defined to describe whether the reciprocal effect of evacuees in different areas exists when they select the same exit to evacuate. The concept of evacuation priority is a group of evacuees that target a common area. Occupants will think about which exit to choose during their evacuation. For example, they will think about the distance to the exit, the possible congestion situation to reach the target exit, etc. Evacuation priority is a factor to consider when they analyze the possible congestion situation because they need to consider who will select the exit for evacuation, thus causing the congestion. Then, they will select an appropriate route to evacuate. As shown in the scenario in Figure 3, if the occupants on the right side select the left exit to evacuate, they need to consider the effects of the occupants in the upper and lower regions. The evacuation priority of the occupants in the upper and lower regions is higher than that of the occupants on the right side. For the evacuees in an area, when the priority of this area is higher than the area of the reference evacuee, the people distribution density of this area is considered as the factor for the evacuation of these evacuees. The evacuation priority is defined as the inverse of the people distribution density, which is described by Equation (2).

$$P_{eva} = 1/P\_dis\_den \tag{2}$$

The total people distribution density ($P\_dis\_den\_sum$) is the sum of people distribution density of all areas in the whole evacuation scene, which is defined by Equation (3).

$$P\_dis\_den\_sum = \sum_k^K \sum_i \frac{N_i^k}{V} \sum_j \frac{L_{ij}^k}{\alpha \cdot d_{ij}^k} \tag{3}$$

where $K$ is the number of areas whose evacuation priority is higher than the position of the reference evacuee.

### 2.3.3. Calculation Example of People Distribution Density

To describe the estimation of people distribution density, an evacuation scenario is constructed using Pathfinder. Pathfinder is an agent-based evacuation simulation developed by the Thunderhead Engineering Company of the United States, which is combined with advanced movement simulation and high-quality 3D animated results [18,53]. Pathfinder enables the analysis for stadiums, hospitals, skyscrapers, aircraft, and other buildings. A scenario is set to give a calculated example of the people distribution density, as shown in Figure 5. There are two exits (exit1 and exit2) and 212 evacuees in this scenario. The average evacuation velocity is set as 1.19 m/s [7]. For the reference evacuee, when exit1 is selected as the evacuation exit, the pedestrians in the three areas including dist1, dist2, and dist4 will have effects on its evacuation.

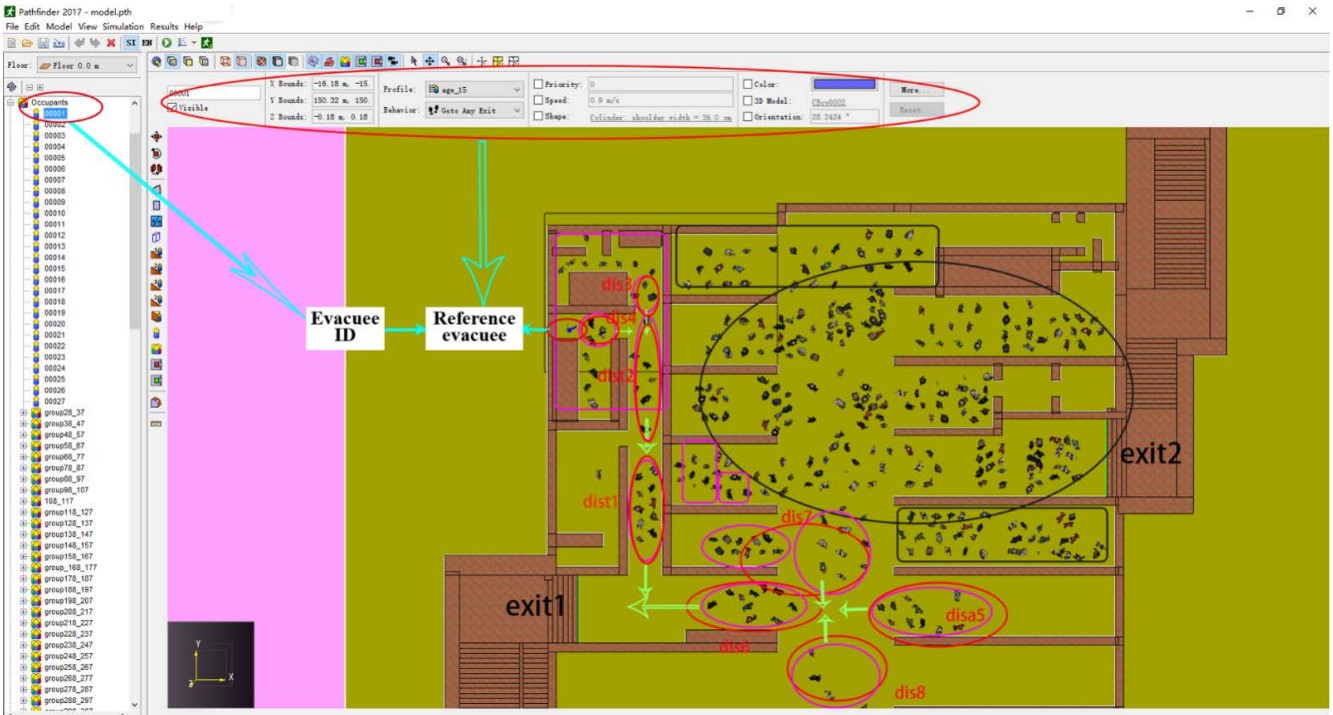

**Figure 5.** An evacuation scenario to describe the estimation of people distribution density.

Formula (4) illustrates the calculation process of the people distribution density for the reference evacuee.

$$P\_dis\_den\_sum = \sum_{k}^{K}\sum_{i}\frac{N_i^k}{V}\sum_{j}\frac{L_{ij}^k}{\alpha \cdot d_{ij}^k}$$

$$= \frac{10}{1.19}\times\left(\frac{4.5}{2.1\times1}+\frac{3}{4\times\frac{3}{4}}\right)+\frac{7}{1.19}\times\left(\frac{11.7}{2.1\times1}+\frac{3}{4\times\frac{3}{4}}\right)+\frac{3}{1.19}\times\left(\frac{2}{1.5\times1}+\frac{14.2}{2.1\times\frac{3}{4}}+\frac{3}{4\times\frac{3}{4}}\right)=93.68$$

(4)

### 2.4. Evacuation Time Estimation Model (ETEM)

2.4.1. Dataset Collection

The factors related to the occupant information are divided into the occupant's shoulder width (*Shoulder_width*), position information (*L*, the distance of the occupant with each emergency exit), the occupant's normal moving speed (V), the occupant's age (age), the occupant's gender (sex), etc. The factors related to congestion degree of the passage are divided into the area width ($d_i$), number of people (*N*), length (*L*) and bending degree (*Channel_diff*) of the evacuation passage. When an evacuation route is not straight, the curvature of the evacuation route is applied to describe its bending degree, which has been divided into four classifications (1, 2, 3) based on the different curved angles.

The underground commercial street of Wanda Plaza in $\times\times$ City is taken as the research object, as shown in Figure 6. The street is about 221 m from north to south and 460 m from east to west. The total construction area is 27,285.38 square meters. The density can reach 3.6 persons per square meter during rush hour. Pathfinder is used to construct the evacuation scenario with 443 occupants and 19 emergency exits. The information of occupants can be obtained through a literature review. The summary of basic statistics information is shown in Table 2.

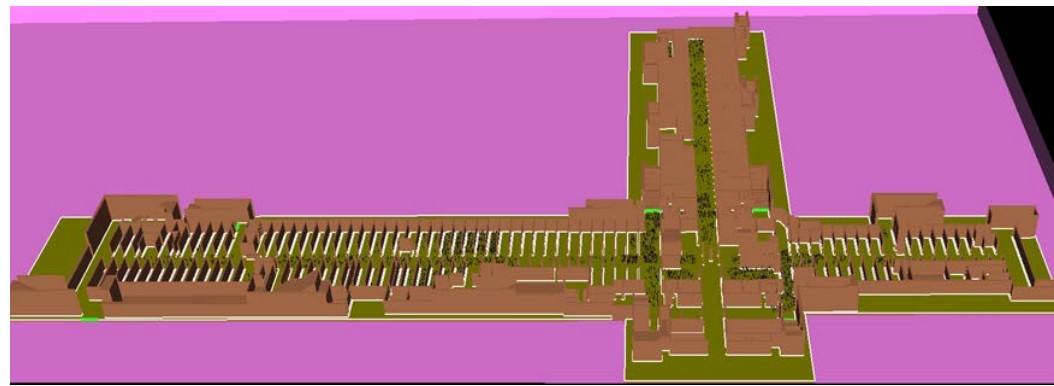

**Figure 6.** The underground commercial street of Wanda Plaza.

**Table 2.** The summary of occupants' basic statistics information.

| Age (Years) | v (m/s) | Sex | *Shoulder_Width* (cm) |
|---|---|---|---|
| Under 15 | 0.9 | 0 | 36 |
| Under 15 | 0.8 | 1 | 32 |
| 16~50 | 1.19 | 0 | 45.58 |
| 16~50 | 1 | 1 | 40 |
| Over 51 | 0.8 | 0 | 45.58 |
| Over 51 | 0.7 | 1 | 40 |

Note: 0 marks male, and 1 marks female.

The data corresponding to factors are recorded automatically in the simulation of the evacuation process through Pathfinder. There are 3443 pieces of simulation data corresponding to the occupants. Partial data are shown in Table 3.

**Table 3.** Partial data recorded by Pathfinder.

| Name | Exit Time (s) | Active Time (s) | Jam Time Total (s) | Jam Time Max Continuous (s) | Start Time (s) | Finish Time (s) | Distance (m) | Last_Goal_Started Time (s) |
|---|---|---|---|---|---|---|---|---|
| 1 | 33 | 33 | 0.35 | 0.33 | 0 | 33 | 24.03 | 0 |
| 2 | 33.93 | 33.93 | 0.65 | 0.33 | 0 | 33.93 | 24.22 | 0 |
| 3 | 38.63 | 38.63 | 0.45 | 0.33 | 0 | 38.63 | 26.61 | 0 |
| 4 | 40.18 | 40.18 | 0.93 | 0.25 | 0 | 40.18 | 26.19 | 0 |
| 5 | 30.95 | 30.95 | 1.58 | 0.95 | 0 | 30.95 | 23.13 | 0 |
| 6 | 33.48 | 33.48 | 1.05 | 0.25 | 0 | 33.48 | 26.2 | 0 |
| 7 | 20.05 | 20.05 | 0.25 | 0.25 | 0 | 20.05 | 20.79 | 0 |
| 8 | 12.93 | 12.93 | 0.25 | 0.25 | 0 | 12.93 | 12.79 | 0 |
| 9 | 13.95 | 13.95 | 0.25 | 0.25 | 0 | 13.95 | 14.82 | 0 |
| 10 | 30.53 | 30.53 | 0.48 | 0.25 | 0 | 30.53 | 24.58 | 0 |
| 11 | 15 | 15 | 0.25 | 0.25 | 0 | 15 | 15.63 | 0 |
| 12 | 44.05 | 44.05 | 0.5 | 0.33 | 0 | 44.05 | 28.70 | 0 |
| 13 | 42.55 | 42.55 | 2.03 | 1.23 | 0 | 42.55 | 26.95 | 0 |
| 14 | 44.98 | 44.98 | 0.45 | 0.35 | 0 | 44.98 | 28.29 | 0 |
| 15 | 39.43 | 39.43 | 1.43 | 0.55 | 0 | 39.43 | 22.91 | 0 |
| 16 | 35.83 | 35.83 | 0.53 | 0.35 | 0 | 35.83 | 22.95 | 0 |
| 17 | 37.9 | 37.9 | 1.63 | 0.75 | 0 | 37.9 | 22.87 | 0 |
| 18 | 39.9 | 39.9 | 0.33 | 0.33 | 0 | 39.9 | 25.87 | 0 |
| 19 | 43.18 | 43.18 | 1.63 | 0.38 | 0 | 43.18 | 27.66 | 0 |
| 20 | 26.68 | 26.68 | 0.28 | 0.28 | 0 | 26.68 | 22.42 | 0 |
| 21 | 31.68 | 31.68 | 2.25 | 1.28 | 0 | 31.68 | 20.72 | 0 |
| 22 | 41 | 41 | 0.4 | 0.3 | 0 | 41 | 28.47 | 0 |

**Table 3.** *Cont.*

| Name | Exit Time (s) | Active Time (s) | Jam Time Total (s) | Jam Time Max Continuous (s) | Start Time (s) | Finish Time (s) | Distance (m) | Last_Goal_Started Time (s) |
|---|---|---|---|---|---|---|---|---|
| 23 | 28.95 | 28.95 | 0.4 | 0.4 | 0 | 28.95 | 19.32 | 0 |
| 24 | 25.7 | 25.7 | 0.48 | 0.4 | 0 | 25.7 | 16.45 | 0 |
| 25 | 25.85 | 25.85 | 0.45 | 0.28 | 0 | 25.85 | 19.27 | 0 |
| 26 | 18.03 | 18.03 | 0.3 | 0.3 | 0 | 18.03 | 17.07 | 0 |
| 27 | 17.35 | 17.35 | 0.53 | 0.3 | 0 | 17.35 | 15.25 | 0 |
| 28 | 11.5 | 11.5 | 0.35 | 0.35 | 0 | 11.5 | 7.82 | 0 |
| 29 | 16.93 | 16.93 | 0.35 | 0.35 | 0 | 16.93 | 12.17 | 0 |
| 30 | 9.33 | 9.33 | 0.35 | 0.35 | 0 | 9.33 | 6.81 | 0 |
| 31 | 12.1 | 12.1 | 0.25 | 0.25 | 0 | 12.1 | 9.17 | 0 |
| 32 | 12.53 | 12.53 | 0.25 | 0.25 | 0 | 12.53 | 10.80 | 0 |
| 33 | 8.68 | 8.68 | 0.25 | 0.25 | 0 | 8.68 | 8.74 | 0 |
| 34 | 13.55 | 13.55 | 0.65 | 0.25 | 0 | 13.55 | 11.82 | 0 |
| 35 | 5.68 | 5.68 | 0.25 | 0.25 | 0 | 5.68 | 5.91 | 0 |
| 36 | 6.9 | 6.9 | 0.25 | 0.25 | 0 | 6.9 | 7.53 | 0 |
| 37 | 9.8 | 9.8 | 0.25 | 0.25 | 0 | 9.8 | 9.73 | 0 |
| 38 | 61.45 | 61.45 | 19.18 | 6.68 | 0 | 61.45 | 34.60 | 0 |

name: the number of evacuation occupants in the simulation; exit time: the evacuation time; active time: the activation time of an evacuation occupant; jam time total: total congestion time; jam time max continuous: the maximum duration of congestion; start time: the start time of evacuation; finish time: the finish time of evacuation; distance: the total distance of evacuation; last_goal_started_time: the time of the last occupant starting the evacuation.

### 2.4.2. Correlation Analysis of Factors

Correlation analysis is the basis of the evacuation time estimation based on these factors. The simulation dataset is applied to build the correlation relationships. The correlation coefficients of different factors are shown in Table 4. The order of each positive correlation factor based on its influence degree is $P\_dis\_den\_sum > distance(m) > channel\_diff > jam\ time\ total(s) > jam\ time\ max\ continuous > exit\_width > sex$. The total people distribution density ($P\_dis\_den\_sum$) is the most significant positive correlation factor, whose coefficient is 0.86. The result also proves that the introduction of this factor is effective for the estimation of evacuation time. The order based on influence degree of each negative correlation factor is $v > shoulder\_width$. The most significant negative correlation factor is velocity ($v$).

**Table 4.** Correlation coefficients.

| | Exit Time | Jam Time Total | Jam Time Max Continuous | Distance | Sex | v | Exit_Width | Channel_Diff | P_Dis_Den_Sum | Shoulder_Width |
|---|---|---|---|---|---|---|---|---|---|---|
| exit time | 1.00 | 0.69 | 0.63 | 0.85 | 0.21 | −0.36 | 0.32 | 0.71 | 0.86 | −0.30 |
| jam time total | 0.69 | 1.00 | 0.97 | 0.28 | 0.043 | 0.15 | 0.27 | 0.40 | 0.43 | −0.042 |
| jam time max continuous | 0.63 | 0.97 | 1.00 | 0.22 | 0.041 | −0.17 | 0.24 | 0.33 | 0.36 | −0.036 |
| distance | 0.85 | 0.28 | 0.22 | 1.00 | 0.16 | −0.17 | 0.8 | 0.71 | 0.87 | −0.25 |
| sex | 0.21 | 0.043 | 0.04 | 0.16 | 1.00 | −0.47 | 0.16 | 0.063 | 0.21 | −0.56 |
| v | −0.36 | −0.15 | −0.17 | −0.17 | −0.47 | 1.00 | −0.17 | −0.082 | −0.24 | 0.65 |
| exit_width | 0.32 | 0.27 | 0.24 | 0.18 | 0.16 | −0.17 | 1.00 | 0.092 | 0.065 | −0.069 |
| channel_diff | 0.71 | 0.40 | 0.33 | 0.71 | 0.063 | −0.082 | 0.092 | 1.00 | 0.73 | −0.11 |
| P_dis_den_sum | 0.86 | 0.43 | 0.36 | 0.87 | 0.21 | −0.24 | 0.065 | 0.73 | 1.00 | −0.32 |
| shoulder_width | −0.30 | −0.043 | −0.036 | −0.25 | −0.56 | 0.65 | −0.069 | −0.10 | −0.31 | 1.00 |

The influence of population distribution density on evacuation time has always been an axiom that is widely believed but has not been verified. It has not been taken into account in the previous study of evacuation time prediction. In this paper, we propose the impact of

population distribution density on evacuation time and turn its impact degree into visual data. The scatter diagram is drawn to describe the correlation between evacuation time and the total people distribution density further, as shown in Figure 7. The figure proves that the correlation is positive. The evacuation time is longer when the density of occupants is higher, which is consistent with common sense.

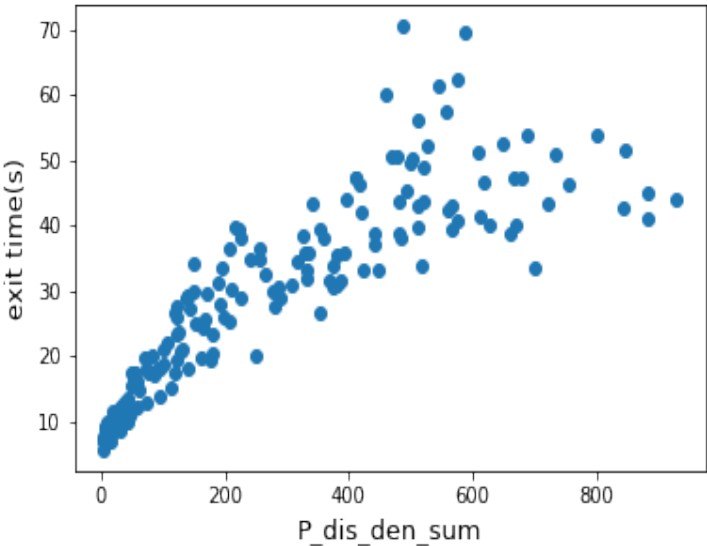

**Figure 7.** The scatter diagram of evacuation time with the total people distribution density.

## 3. Results

A machine learning model is established to conduct the experiments on the simulation dataset. The prediction accuracy of the model is verified by the testing set. In this paper, the stacking strategy with five-fold cross-validation is used through integrating classifiers of LGB, XGB and GBoost. Cross-entropy is an important concept in Shannon's information theory, which is mainly used to measure the difference information between two probability distributions [54]. The final cross-entropy on the training set is 3.46, and the variance is 0.83. On the testing set, the mean error between the predicted evacuation time with the model and the actual simulated time is 3.63 s.

### 3.1. Data Normalization

The data related to some factors cannot be recorded by Pathfinder, such as total people distribution density (*P_dis_den_sum*), bending degree (*channel_dif*), the width of the route (*exit_width*), velocity (*v*), and age. These data are added into the dataset manually recorded by the simulation in the evacuation process. The partial complete dataset is shown in Appendix A Table A1.

Data normalization is the basis for machine learning to avoid overfitting and bias and to improve accuracy. Data normalization is calculated by Equation (5). Some samples of data normalization are represented in Table 5.

$$x_i = \frac{x_i - \mu}{\sigma} \tag{5}$$

where $x_i$ is the value of a certain factor for an occupant (*i*), $\mu$ is the average value of this factor, and $\sigma$ is its mean square deviation. The evacuation time (*exit time(s)*) is not normalized to compare the predicted value with the simulation result more directly.

**Table 5.** Partial normalized data.

|  | Exit Time | Distance | Sex | v | Exit_Width | Channel_Diff | P_Dis_Den_Sum | Shoulder_Width |
|---|---|---|---|---|---|---|---|---|
| 0 | 33.00 | 0.83 | −0.92 | −0.53 | −1.53 | 0.89 | 0.68 | −1.13 |
| 1 | 33.93 | 0.86 | −0.92 | −0.53 | −1.53 | 0.89 | 0.97 | −1.13 |
| 2 | 38.63 | 1.24 | −0.92 | −0.53 | −1.53 | 0.89 | 1.58 | −1.13 |
| 3 | 40.18 | 1.17 | −0.92 | 1.13 | −1.53 | 2.05 | 1.44 | 0.90 |
| 4 | 30.95 | 0.69 | −0.92 | 1.13 | −1.53 | 0.89 | 0.40 | 0.90 |
| 5 | 33.48 | 1.17 | −0.92 | 1.13 | −1.53 | 0.89 | 1.74 | 0.90 |
| 6 | 20.05 | 0.32 | −0.92 | 1.13 | −1.53 | 0.89 | −0.15 | 0.90 |
| 7 | 12.93 | −0.94 | −0.92 | 1.13 | −1.53 | −0.27 | −0.89 | 0.90 |
| 8 | 13.95 | −0.62 | −0.92 | 1.13 | −1.53 | −0.27 | −0.81 | 0.90 |
| 9 | 30.53 | 0.92 | −0.92 | 1.13 | −1.53 | 0.89 | 0.38 | 0.90 |

*3.2. Evacuation Time Prediction*

The model is implemented by the Anaconda machine learning platform. The correlation analysis among the factors in Section 2.4.2 is the basis for constructing an evacuation time prediction model. The function is described as Equation (6).

$$y\_pred\_i = \sum_{t=0}^{T} \alpha_t h_t(x_i; w_t) \tag{6}$$

where $y\_pred\_i$ is the predicted evacuation time for an occupant ($i$), $h_t$ is the $t_{th}$ classifier ($T$ is the total number of classifiers), and $\alpha_t$ and $w_t$ represent the weight and parameters of the classifier, respectively.

There are 300 occupants in the training set. $y\_sim\_i$ is the simulation time by Pathfinder, which is considered as the reference value to calculate the loss value. The loss function is the mean square error (mse), as shown in Equation (7).

$$Loss = \frac{1}{N} \sum_{i=1}^{i=N} \sqrt{(y\_pred\_i - y\_sim\_i)^2} \tag{7}$$

Three classifiers are selected, which are XGB, LGB, and GBoost. The training process is updated based on Equation (6). Loss values for the three classifiers are 3.709, 3.718, and 3.963 s, respectively.

*3.3. Validation of the Prediction Model*

There are 60 samples in the simulated dataset for model validation. The error value is the mean square error between the predicted evacuation time and the simulated evacuation time. The error on the testing set is 3.63 s, which is slightly larger than the training error of 3.41 s. The result shows slight overfitting, which can be ignored when the number of samples is small. The loss is 3.42 s in the integration model, which is shorter than any single classifier.

The main causes of casualties usually are wrong route selection and difficulty in evacuation. In the process of fire evacuation, the evacuees are racing against time. Once the selected evacuation route takes a long time, they are very likely to be choked by smoke. Therefore, the route that fails to evacuate successfully and causes casualties is the wrong route. Difficulty in evacuation refers to the passengers feeling as if it is more difficult than the normal channel in the evacuation process due to some factors such as complex design of the evacuation passage and obstacles in the channel. The model can predict the evacuation time consumed by occupants in advance based on the combination of individual factor and dynamic environmental information. At the same time, experts such as the designers and builders of the building can receive help from the evacuation time estimation to make the building more scientific and safer. When the model is embedded into the intelligent guidance system, the evacuation time can be predicted accurately. Then, occupants can plan the evacuation route with the shortest time by being told from the warning materials such as

maps or signs, and the faster route can be presented and sent to evacuees in real time through the guidance system. Finally, the overall evacuation efficiency can be improved greatly.

## 4. Discussion and Conclusions

Based on the correlation between a variety of factors and evacuation time, a prediction model is proposed using stacking integration strategy. The main value of our study to Pathfinder is the introduction of people distribution density. Then, the influence of unstable pedestrian flow and unbalanced distribution can be considered into the estimation of evacuation time. The main conclusions are as follows:

1.  The influencing factors of evacuation time are analyzed from two categories: user information and channel congestion situation. To consider the influence of unstable pedestrian flow and unbalanced distribution, the concept of people distribution density is introduced to the evacuation time prediction model. Based on the typical scenario analysis, the definition and estimation method are proposed. The Pathfinder model is applied for evacuation simulation to create the dataset. Then, the simulation data are recorded and standardized to create machine learning datasets.
2.  The correlation analysis is conducted to assess the effect of each factor on evacuation time. The correlation coefficient of people distribution density on evacuation time is 0.86. The results show that the total people distribution density is the most significant positive correlation factor. Its introduction can effectively improve the prediction accuracy of our model.
3.  The evacuation time prediction model is put forward and implemented by the Anaconda machine learning platform. After learning the relationship between each factor and evacuation time in the training dataset, the model can predict evacuation time in advance when the occupants are preparing to evacuate. Compared with the actual evacuation time, the average error of the predicted time is 3.63 s, which proves that our model can support more accurate route planning for emergency situations.

There are some limitations of our research. In the study of evacuation time prediction, normally, it should be verified by evacuation experiments. However, some restrictions exist in our study. For example, it is very difficult to conduct evacuation experiments on large entertainment venues. Therefore, the simulation time of Pathfinder is simply set as the reference value for model verification. Furthermore, the model has some shortcomings. It is only applicable to the fire evacuation prediction of large buildings. Nevertheless, there are many types of large buildings in reality, but only part of the fire evacuation of large buildings may be consistent with the model we established. Moreover, there are many factors influencing evacuation efficiency, but we mainly focused on the dynamic information of individual factors and congestion degree in this paper. Some other factors may not be considered in this study. Although it is not very comprehensive, this study brings new insight into the fire evacuation research of large buildings when dynamic information is considered in the evacuation process. The model can be used to predict the evacuation time of each route for users more accurately. At the same time, the most appropriate evacuation path can be provided for the evacuation selection of each floor according to the proposed people distribution density. It is conducive to providing effective guidance for evacuation plans of large buildings and minimizing casualties. In future research, we will incorporate more factors from a real evacuation scene, so that our model can be more capable of solving the problem of large building evacuation. A more comprehensive model will be considered in our intelligent evacuation guidance system in future research.

**Author Contributions:** Conceptualization, Q.D., B.Z., L.Z. and H.J.; Methodology, Q.D. and B.Z.; Formal analysis, Q.D. and Z.Z. (Zheng Zhou); Investigation, Q.D. and B.Z.; Data curation, Q.D. and Z.Z. (Zhengqing Zhou); Writing—original draft, Q.D., Z.Z. (Zheng Zhou) and H.D.; Writing—review & editing, Q.D., H.D., L.Z., Z.Z. (Zhengqing Zhou) and H.J.; Visualization, Q.D., B.Z. and H.D.; Supervision, H.J.; Funding acquisition, Q.D., L.Z., Z.Z. (Zhengqing Zhou) and H.J. All authors have read and agreed to the published version of the manuscript.

**Funding:** This research was funded by the National Key R&D Program of China (No. 2021YFC1523504), National Science Foundation of China (grant Nos. 72004113, 72174099, 72104123, 71904193), Science and Technology Program of the Ministry of Emergency Management (No. 2021XFCX25), and High-tech Discipline Construction Fundings for Universities in Beijing (Safety Science and Engineering).

**Institutional Review Board Statement:** Not applicable.

**Informed Consent Statement:** Not applicable.

**Data Availability Statement:** The authors declare that the data supporting the findings of this study are available within the article.

**Conflicts of Interest:** The authors declare that there is no conflict of interest regarding the publication of this article.

## Abbreviations

List of symbols and abbreviations.

| | |
|---|---|
| $T_{ASE}$ | available evacuation time |
| $T_{RSE}$ | required evacuation time |
| $T_{EVA}$ | evacuation time |
| $T_{MOV}$ | movement time |
| $T_{WAIT}$ | waiting time |
| $T_{AWA}$ | awareness time |
| $T_{PRE}$ | pre-movement time |
| **GA** | genetic algorithm |
| **NNs** | neural networks |
| **FL** | fuzzy logic |
| **ANN** | artificial neural network |
| **ETEM** | evacuation time estimation model |
| **XGB** | extreme gradient boosting |
| **LGB** | light gradient boosting |
| **GBoost** | gradient boosting |
| **mse** | mean square error |
| **exit time (s)** | evacuation time |
| **active time (s)** | activation time of an evacuation occupant |
| **jam time total (s)** | total congestion time |
| **jam time max continuous (s)** | the maximum duration of congestion |
| **start time (s)** | start time of evacuation |
| **finish time (s)** | finish time of evacuation |
| **distance (m)** | total distance of evacuation |
| **last_goal_started time (s)** | time of the last occupant starting evacuation |
| **sex (years)** | occupant's gender |
| **v (m/s)** | occupant's normal moving speed |
| *exit_width* **(m)** | width of the exit |
| *channel_diff* | bending degree |
| *P_dis_den_sum* | total people distribution density |
| *shoulder_width* **(cm)** | occupant's shoulder width |
| **age (years)** | occupant's age |

## Appendix A

Table A1. Partial complete dataset.

| Name | Exit Time (s) | Active Time (s) | Jam Time Total (s) | Jam Time Max Continuous (s) | Start Time (s) | Finish Time (s) | Distance (m) | Last_Goal_ Started Time (s) | Sex | V | Exit_Width (m) | Channel_Diff | P_Dis_Den _Sum | Shoulder _Width (cm) |
|---|---|---|---|---|---|---|---|---|---|---|---|---|---|---|
| 1 | 33 | 33 | 0.35 | 0.33 | 0 | 33 | 24.03 | 0 | 0 | 0.9 | 410 | 2 | 447.25 | 36 |
| 2 | 33.93 | 33.93 | 0.65 | 0.33 | 0 | 33.93 | 24.22 | 0 | 0 | 0.9 | 410 | 2 | 516.25 | 36 |
| 3 | 38.63 | 38.63 | 0.45 | 0.33 | 0 | 38.63 | 26.61 | 0 | 0 | 0.9 | 410 | 2 | 660.25 | 36 |
| 4 | 30.95 | 30.95 | 1.58 | 0.95 | 0 | 30.95 | 23.13 | 0 | 0 | 1.2 | 410 | 2 | 381.25 | 45.58 |
| 5 | 33.48 | 33.48 | 1.05 | 0.25 | 0 | 33.48 | 26.2 | 0 | 0 | 1.2 | 410 | 2 | 699 | 45.58 |
| 6 | 20.05 | 20.05 | 0.25 | 0.25 | 0 | 20.05 | 20.79 | 0 | 0 | 1.2 | 410 | 2 | 249.8 | 45.58 |
| 7 | 12.93 | 12.93 | 0.25 | 0.25 | 0 | 12.93 | 12.79 | 0 | 0 | 1.2 | 410 | 1 | 74.5 | 45.58 |
| 8 | 13.95 | 13.95 | 0.25 | 0.25 | 0 | 13.95 | 14.82 | 0 | 0 | 1.2 | 410 | 1 | 94.5 | 45.58 |
| 9 | 30.53 | 30.53 | 0.48 | 0.25 | 0 | 30.53 | 24.58 | 0 | 0 | 1.2 | 410 | 2 | 376 | 45.58 |
| 10 | 15 | 15 | 0.25 | 0.25 | 0 | 15 | 15.63 | 0 | 0 | 1.2 | 410 | 2 | 114.5 | 45.58 |
| 11 | 44.05 | 44.05 | 0.5 | 0.33 | 0 | 44.05 | 28.70 | 0 | 1 | 0.8 | 410 | 2 | 929.35 | 32 |
| 12 | 42.55 | 42.55 | 2.03 | 1.23 | 0 | 42.55 | 26.95 | 0 | 1 | 0.8 | 410 | 2 | 842.65 | 32 |
| 13 | 44.98 | 44.98 | 0.45 | 0.35 | 0 | 44.98 | 28.29 | 0 | 1 | 0.8 | 410 | 3 | 883.75 | 32 |
| 14 | 39.43 | 39.43 | 1.43 | 0.55 | 0 | 39.43 | 22.91 | 0 | 1 | 0.8 | 410 | 1 | 566.25 | 32 |
| 15 | 35.83 | 35.83 | 0.53 | 0.35 | 0 | 35.83 | 22.95 | 0 | 1 | 0.8 | 410 | 1 | 392.05 | 32 |
| 16 | 37.9 | 37.9 | 1.63 | 0.75 | 0 | 37.9 | 22.87 | 0 | 1 | 0.8 | 410 | 1 | 483.25 | 32 |
| 17 | 39.9 | 39.9 | 0.33 | 0.33 | 0 | 39.9 | 25.87 | 0 | 1 | 0.8 | 410 | 2 | 668.95 | 32 |
| 18 | 43.18 | 43.18 | 1.63 | 0.38 | 0 | 43.18 | 27.66 | 0 | 1 | 0.8 | 410 | 2 | 720.25 | 32 |
| 19 | 26.68 | 26.68 | 0.28 | 0.28 | 0 | 26.68 | 22.42 | 0 | 1 | 1 | 410 | 1 | 352.45 | 40 |
| 20 | 31.68 | 31.68 | 2.25 | 1.28 | 0 | 31.68 | 20.72 | 0 | 1 | 1 | 410 | 1 | 368.53 | 40 |
| 21 | 41 | 41 | 0.4 | 0.3 | 0 | 41 | 28.47 | 0 | 1 | 1 | 410 | 2 | 881.25 | 40 |

**Table A1.** *Cont.*

| Name | Exit Time (s) | Active Time (s) | Jam Time Total (s) | Jam Time Max Continuous (s) | Start Time (s) | Finish Time (s) | Distance (m) | Last_Goal_ Started Time (s) | Sex | V | Exit_Width (m) | Channel_Diff | P_Dis_Den _Sum | Shoulder _Width (cm) |
|---|---|---|---|---|---|---|---|---|---|---|---|---|---|---|
| 22 | 28.95 | 28.95 | 0.4 | 0.4 | 0 | 28.95 | 19.32 | 0 | 1 | 0.7 | 410 | 1 | 224.45 | 40 |
| 23 | 25.7 | 25.7 | 0.48 | 0.4 | 0 | 25.7 | 16.45 | 0 | 1 | 0.7 | 410 | 1 | 167 | 40 |
| 24 | 25.85 | 25.85 | 0.45 | 0.28 | 0 | 25.85 | 19.27 | 0 | 1 | 1 | 410 | 1 | 198.5 | 40 |
| 25 | 18.03 | 18.03 | 0.3 | 0.3 | 0 | 18.03 | 17.07 | 0 | 1 | 1 | 410 | 1 | 139.5 | 40 |
| 26 | 17.35 | 17.35 | 0.53 | 0.3 | 0 | 17.35 | 15.25 | 0 | 1 | 1 | 410 | 1 | 118.5 | 40 |
| 27 | 11.5 | 11.5 | 0.35 | 0.35 | 0 | 11.5 | 7.82 | 0 | 0 | 0.8 | 410 | 1 | 17.6 | 45.58 |
| 28 | 16.93 | 16.93 | 0.35 | 0.35 | 0 | 16.93 | 12.17 | 0 | 0 | 0.8 | 410 | 1 | 52 | 45.58 |
| 29 | 9.33 | 9.33 | 0.35 | 0.35 | 0 | 9.33 | 6.81 | 0 | 0 | 0.8 | 410 | 1 | 8.4 | 45.58 |
| 30 | 12.1 | 12.1 | 0.25 | 0.25 | 0 | 12.1 | 9.17 | 0 | 0 | 1.2 | 410 | 1 | 30.8 | 45.58 |
| 31 | 12.53 | 12.53 | 0.25 | 0.25 | 0 | 12.53 | 10.80 | 0 | 0 | 1.2 | 410 | 1 | 37 | 45.58 |
| 32 | 8.68 | 8.68 | 0.25 | 0.25 | 0 | 8.68 | 8.74 | 0 | 0 | 1.2 | 410 | 1 | 13.7 | 45.58 |
| 33 | 13.55 | 13.55 | 0.65 | 0.25 | 0 | 13.55 | 11.82 | 0 | 0 | 1.2 | 410 | 1 | 44 | 45.58 |
| 34 | 5.68 | 5.68 | 0.25 | 0.25 | 0 | 5.68 | 5.91 | 0 | 0 | 1.2 | 410 | 1 | 4 | 45.58 |
| 35 | 6.9 | 6.9 | 0.25 | 0.25 | 0 | 6.9 | 7.53 | 0 | 0 | 1.2 | 410 | 1 | 5 | 45.58 |
| 36 | 9.8 | 9.8 | 0.25 | 0.25 | 0 | 9.8 | 9.73 | 0 | 0 | 1.2 | 410 | 1 | 25.6 | 45.58 |

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
