# Peer review of "Evacuation Time Estimation Model in Large Buildings Based on Individual Characteristics and Real-Time Congestion Situation of Evacuation Exit"

_fire, doi:10.3390/fire5060204_

Round 1
Reviewer 1 Report
GENERAL COMMENTS
The objective of this paper is very unclear. The authors have conducted some simulations using Pathfinder including an underground commercial street (line 308), a seemingly arbitrary configuration (Figure 5) and a 300-agent training set (line 386). It is confusing to the reader why these three simulation scenarios are necessary. The authors then present what they appear to call as their ‘Evacuation time estimation model’ which they proceed to do some kind of calculation to get the ‘people distribution density’. It is not clear to me how this value is a density (i.e., persons per metre squared in evacuation terminology) and also where numbers are derived from in their example. Following this there is an application of ‘stacking integration’, which seems to combine ‘people distribution density’ with some other factors which either come from Pathfinder or need to be determined separately. The authors then suggest that knowing the ‘people distribution density’ allows the best determination of each agent’s total evacuation time because of the strong correlation between the two. The final argument about ‘people distribution density’ being the best is not too convincing given the very similar correlation between distance and evacuation time. However, the purpose of all of this is unclear – if the desire is to get the agent evacuation times then those are already given by Pathfinder. The authors appear to suggest that “…the model can help occupants to plan the evacuation route with the shortest time” but I think this is a wholly unrealistic expectation as how will occupants be able to access such information, and would they really be considering an evacuation planning as part of their daily life? – I very much doubt it. Even if they were, how would occupants know where other people were, how would they know information such as average walking speeds etc.? The authors then suggest that “When the model is embedded into the intelligent guidance system, the evacuation time can be predicted accurately” but this has not been demonstrated in the paper so how do the authors know this is the case? Again, how would such a system know average walking speeds, the exit choice that an occupant might make etc. to then somehow direct people.
To illustrate the lack of clarity in the paper I have had to make a number of comments just on the abstract before even getting to the main part of the paper.
Although the paper is sufficiently readable it would benefit from some editorial input from a native English speaker to improve the text.
Supposedly this paper is for the special issue on ‘Turbulent combustion fires’ but this paper has nothing to do with that topic.
DETAILED COMMENTS
Comments are preceded by the relevant line number.
15: The abstract claims that “Fire is one of the most common and harmful disaster in real life. People are easy to be panic and make irrational decisions in fire situations.” How do the authors measure what is the ‘most common and harmful’ disasters are? More importantly the authors need to read the evacuation and human behaviour literature before claiming “People are easy to be panic” as this an unsupported claim.
17: What is meant by “…support for intelligent evacuation”? Intelligent evacuation by occupants?
18: What is meant by an ‘intricate evacuation channel’. It is possible this has not been expressed well in English because I do not understand what a ‘channel’ is, or what makes it ‘intricate’ in this context.
21: The abstract states “…unstable pedestrian flow and unbalanced distribution in the process of evacuation” but it is very unclear what is meant by ‘unstable flow’ and also ‘unbalanced distribution’. For example, does unbalanced distribution mean that occupants are not uniformly distributed around a space at the start of an evacuation, or that the choice of exits is unbalanced? (and if the second case how is unbalanced defined – by numbers, by numbers per unit width, some other measure?).
24: It is stated that “…Pathfinder to learn the relationship between evacuation time and influencing factors” but it is unclear how Pathfinder ‘learns’, or maybe this sentence has not been expressed very clearly.
26: What is “consumption time”?
42: The authors suggest that the main reason that fatalities occurred in the Cook Country fire was because “…occupants [were] failed to evacuate in time” but as I recall it was not as simple as this and part of the reason was their choice of exit. I am not sure that picking on two specific fires for the paper adds much value but instead raises questions.
47: The paper states “The occupants inside the fire site only have a few minutes to evacuate…” but this is a very sweeping statement as there are many factors such as the nature of the fire, the location of occupants in relation to the fire etc. which might mean they have a lot longer than just a few minutes
74: Table 1 and the text on page 3 and 4 is a long list of papers that the authors have seen but add very little to the current manuscript. It makes some statements which are very unclear such as “…pre-action time and the stability region…” and “…an occupant has stronger escape ability, and the escape environment is better, i.e., the evacuation route is loose…”. The section discusses construction site fire management, mine fires, platform levels, earthquake casualties all of which are not relevant to “large shopping centers and entertainment venues” introduced on line 36. The introductory text needs to focus on those topics specifically relevant to this paper rather than trying to be a catch-all discussion.
190: The discussion here is in the first person (i.e. ‘you’) but this should be written in the third person (i.e. ‘a person’).
205: The text that states “…once there are obstacles in the evacuation space that cannot be crossed over, pedestrians will avoid bypassing the obstacles and pass through them at a lower speed”. This is confusing as what is meant by an obstacle that cannot be ‘crossed over’ but then can be passed through at a lower speed?
207: When the paper states “pedestrians will return to the scene of the fire again because of various behaviors such as saving lives, putting out fires, and taking valuables” are the authors proposing that their analysis will incorporate these factors – if not then why mention it?
213: I agree that “Therefore, the real evacuation scenario is very complex” and models such as Pathfinder do not address a number of the factors previously listed, and even where it does it is unclear where and how the authors have applied these in their analysis. The paper states that “People distribution density is introduced to illustrate the coupling effect of these dynamic behavior” but I cannot see where it includes factors such as waiting for family groups, returning to collect valuables etc.
219: The paper suggests that “One of main reasons that evacuees choose wrong routes…” but it is unclear why a route is defined as ‘wrong’ – for example if an occupant wants to go the way they came in then that route is not ‘wrong’ in their eyes irrespective of how crowded it might be. Maybe their choice is not optimal for total evacuation time, but that is not ‘wrong’ as such.
264: “is the effective coefficient of width” seems to have something missing at the start.
265: “According to a lot of observations…”, please provide references to support this statement.
270: It is unclear what is meant by the ‘evacuation priority’ and how it has been found to be of relevance to the analysis. It seems to be applied to each agent and seems to be when “they select the same exit to evacuate” so does that mean each agent needs to know which exit every other agent is going to. If so, how would that work in reality as a person would never know which exit all the other people were going to.
285: Explain in what way Pathfinder is an “intelligent evacuation evaluation system”? Also, I am not convinced it works “…under disaster conditions” – how does it account for a ‘disaster’?
289: Explain how is “The average evacuation velocity is set as 0.9” (I assume this is in m/s) since the velocity of the agents will vary as they encounter congestion so the average cannot be guaranteed.
291: Since the model uses agents that are not real people then I suggest referring to them as ‘its’ e.g. “will have effects on its evacuation.”
299: Where do the numbers in the formula come from and why is it apparently just the sum of three people distribution densities?
308: It is unclear what the purpose of introducing the Wanda Plaza is to the paper. The configuration seems to be just two long streets but supposedly with 19 exits but how this is modelled is not described in a form the reader can follow. Previously the paper has discussed ‘intricate evacuation channels’ but this configuration does not appear to be ‘intricate’.
311: The paper states “The evacuation of high-density occupants is a [c]hallenging problem in emergencies.” I am not sure the occupants have a ‘high-density’ so maybe this should be ‘spaces that are have a high density of occupants’ but why is this a challenging problem in emergencies and how does the method proposed in this paper assist?
312: If the work has located 443 occupants in 27285 square meters then how is this deemed to have a high density? It is nowhere near the quoted 3.6 persons per square meter.
315: Table 2, why are those over 51 only male? Where do the values for V and shoulder_width come from? What percentage of each agent type are involved in the simulations? Why specify these parameters separately for the analysis as they are not all independent variables – e.g., velocity is a function of age and sex. How does it make sense to assign a number to sex – what if someone defined (say) +10 for female and -10 for male would that change the analysis?
322: Table 3, it seems ‘exit time’ is the same as ‘finish time’ and ‘active time’ so why show these separately. The ‘start time’ and the ‘last_goal_started time’ (whatever that is) are always zero so why show these in the table. If the start time is 0 then has pre-evacuation delay been ignored? If so, how is ignoring this justified especially given the introduction discusses this factor? Overall, I am not sure what the inclusion of Table 3 to the paper adds to the understanding of the work.
341: Table 4. It is quite hard to get a sense of where the maximum and minimum values are from a large number of values. Since the correlation are symmetrical it is not necessary to show half of the results. Even so, on line 336 the authors claim that “The total people distribution density (?_???_???_???) is the most significant positive correlation factor, whose coefficient is 0.86” and this is for exit time (s) so is the rest of the table needed? Anyway, distance (m) has a positive corelation of 0.85 so the difference between 0.85 and 0.86 is marginal and this would suggest distance would also be a sufficient measure of exit time without then need to somehow determine ‘total people distribution density’.
361: What exactly is the “curvature of the evacuation route (?ℎ?????_???)”? It is also unclear what is the width of an exit route since the width may vary at different places.
366: I do not see a great deal of value in Figure 7 – how is the reader supposed to use a screen grab from a spreadsheet? It would seem this is a repeat of much of Table 3 with some additional columns of numbers. For some unexplained reason Item 34 is labelled as ‘abnormal’ but there is no explanation.
377: Table 5, what is S_C_dis? Why show sex and exit_width as they are constant, why is sex constant when it varies between male and female?
398: The paper states “The loss is 3.4168s in the integration model, which is shorter than any single learner”. Is it meaningful to quote a time to four decimal places here (I am not even sure two decimal places has much meaning when discussing the error as being 3.63 s). In addition, what is meant by the ‘learner’ here – the research is using agents in a simulation which do not have an ability to ‘learn’.
400: The paper states “In the actual evacuation process, the evacuation time will be greatly increased, and even stampedes will occur due to the overcrowding of a passage in crowded places.” It is very unclear what the authors are saying here – if the actual evacuation time will greatly increase what is the objective of doing any of the proposed analysis? I am even less clear why the topic of stampedes has been raised – if nothing else tools such as Pathfinder do not address stampedes.
401: The authors claim that “The main causes of casualties usually are wrong route selection and difficulty in evacuation.” There is no information to support this statement – why does a casualty occur because a person chooses the ‘wrong’ route. How is ‘wrong’ defined here? What is meant by ‘difficulty in evacuation’?
428: This is the first place in the paper there is reference to the Anaconda machine learning platform so I do not understand its relevance or what it even is.
Author Response
The response to comments is in the Word.

Reviewer 2 Report
The manuscript describes an evacuation time prediction model for large buildings. It is stated that the model can assist building occupants in choosing the most favorable channel for evacuation in advance.
I greatly appreciate your efforts for improving safe evacuation which is of crucial importance in fire incidents in large buildings. However, I have some remarks to further develop the manuscript.
1) The manuscript would benefit of language checking by a native English speaker. The text is mostly understandable, but there are some strange expressions, e.g., in the abstract “People are easy to be panic and make irrational decisions …”.
· 2) The manuscript would also benefit of careful proofreading to eliminate typos.
· 3) Some self-evident issues regarding safe evacuation are stated, such as “The main reason in these cases is that the occupants were failed to evacuate in time” in the Introduction. Avoid self-evidences, please.
· 4) You should add a list of symbols and abbreviations. There are several figures, tables and equations showing various quantities which are not always explained within the close context. Pay also attention to showing the units of the quantities, please.
· 5) Figure 1: Explain “Decision to act”, please.
· 6) Table 1, Movement time: The gender and number of occupants are not the only influencing factors. How do you take into account the age and possible disabilities (movement such as crutches or wheelchair; sensory disabilities such as sight or hearing defect; mental disorders)?
· 7) Table 1, Available safe evacuation time: What about radiant heat flux in addition to the factors listed?
· 8) How do you take into account the familiarity of the premises to the occupants? It is different for people in a shopping centre, an office building, an industrial building, a mine, etc. That is, whether the occupants are there repeatedly or occasionally.
· 9) Section 2.2: How do the effect of pollutant concentration, house price, and medical data integration in disease prediction relate to your study of evacuation time prediction?
· 10) Page 8/17, lines 265-268: “According to a lot of observations and simulation experiments, the effective width …”. Any references?
· 11) Page 8/17, line 289-290: “The average evacuation velocity is set as 0.9.” What does this mean? Does it have a unit or is it a proportional value to something?
· 12) Page 9/17, equation 4: It is difficult to follow since the terms in the equation and the numerical example are in different order (alpha etc.).
· 13) Page 9/16, line 310: “The total construction area is 27285.28 square meters.” Can you really give the area within this accuracy? I think that 27285 square meters would be the most accurate possible, taking into account measurement inaccuracy.
· 14) Line 311: challenging instead of hallenging
· 15) Lines 313-314: literature instead of iterature
· 16) Table 2: Give units: Age (years), Shoulder_width (cm). How can you give some of the shoulder widths with the accuracy of 0.01 cm? This is unrealistic. Why women over 51 years are missing?
· 17) Table 3: The presented accuracy of data is unreasonable: times with 0.01 s, distance with 0.01 m = 1 cm. I understand that the values come from simulations, but this kind of accuracy of values does not have any relation to reality. Accuracy of 1 s and 1 m would be more realistic.
· 18) Page 11/17, lines 335-339: I don’t understand what you are trying to say on these lines. Clarify, please.
· 19) Page 11/17, lines 345-346: “The evacuation time will be longer when the density of occupants is higher, which is consistent with the common sense.” I fully agree. Explain what is the additional value of your study from this viewpoint, please.
· 20) Page 12/17, lines 355-357: “The final cross-entropy score …” Explain the meaning of these sentences better, please. Also consider if the giving four decimals for the score and variance and 0.01 s for the mean error are reasonable or too high accuracies, please.
· 21) Page 13/17, line 381: You refer to Section 4.2 which does not exist. Check, please.
· 22) Page 14/17, lines 405-406: “…, the model can help occupants to plan the evacuation route with the shortest time.” How is this taken to practice? The occupants hardly use the model themselves.
· 23) Section 4. Discussion and Conclusions: Explain what is the additional value of your study to Pathfinder, please. It remains slightly unclear. Furthermore, it is a fact that evacuation time predictions should be experimentally verified. What is your plan for that?
Author Response

(The authors gave the same response as above.)

Reviewer 3 Report
This paper attempted to effectively improve the evacuation efficiency of people in large buildings, especially for the scenario with intricate evacuation channels. Considering the complexity of individual behavior, occupant density and exit width, this paper may have some guiding significance for evacuation time prediction. The authors introduced the people distribution density to effectively assess the impact of unstable pedestrian flow and unbalanced distribution in the process of evacuation. An evacuation time prediction model is established to estimate the consumption time of people based on Stacking integration. Some interesting findings are drawn. Several comments are made as follows.
First, there are several grammatical or language errors throughout the manuscript. It should be carefully checked before publication.
Some detailed comments:
1. Section 2.3, more information about the people distribution density must be given if possible. Why the people distribution density will change? What is the concrete interfering factor of the people distribution density? Further more, whether environmental factors will also affect the changes of the people distribution density in addition to human behavior?
2. I think the Section 2.4 should be in the results Section.
3. Section 2.3.1, examples of evacuation routes sound interesting. A diagram illustrating this setup may be valuable. These two cases cannot represent the actual evacuation routes. Whether more situations need to be considered to make the study of evacuation routes impact more comprehensive?
4. More discussions should be added, such as the advantages and disadvantages of the evacuation time prediction model, the scenarios to which the model fits.
5. The Conclusion Section should be revised further to summarize the full manuscript more precisely.
6. For Figure 7, it is better to show it in the form of a table rather than a figure.
Author Response

(The authors gave the same response as above.)

Reviewer 4 Report
Detail remarks for specific text lines:
15 and 35 – “People are easy to be 15 panic and make irrational decisions” – unproven statement. Overall, it is the opposite.
76 to 101 – This part is very chaotic. What is purpose of it? Clarification is needed.
125 – Overall, the Introduction is very long and doesn’t describe any details about the paper.
156 – XGB, LGB and GBoost – need more explanations
162 – Figure 2. Why Phase 1. does not influence the rest of the phases?
165 – is or was proposed?
194 -196 and 223 – Please check grammar
264 – α is missing
265 – some basis for assumption is needed
311 – “hallenging”?
312 – no visualization of the research object
The proposed paper does not give any visualizations for the modelled area. Thus, it is not clear what influence can architecture have on evacuation times. It is also not clear if there is a need for computer simulations for every area. If so, there might be little help from any machine learning because there is now possibility to check people density in real time – discussion is also needed here. There is no discussion on bottle neck effects, which typically have biggest influence on required evacuation time.
Regarding the simulation results (and not knowing the actual layout) – the evacuation times are very short. The jam times are insignificant. It looks like this research should caried out on some more difficult object.
Probably the proposed methodology is very interesting and innovatory, but it must be tested on more challenging area with more bottlenecks and with actual influence on evacuation direction. The evacuation directions are described as a functionality in the paper but the possibility of connection between the proposed algorithms and actual evacuation signage and it’s consequences is not described.
Author Response

(The authors gave the same response as above.)

Round 2
Reviewer 1 Report
The responses to my comments and what exactly has changed in the revised paper is not very clear. I was expecting to see specific revisions to the text as tracked changes and also those changes to be written out in the response from the author/s. Given this has not been done it is hard to assess the revised paper. Simply highlighting parts of the revised paper and saying 'that content has been added' is not particularly helpful for me as a reviewer.
It seems that the authors have addressed several of my minor points, but they have not really made an attempt to consider the major comments regarding the utility of the approach given it seems knowing the distance to the exit is all that is necessary and how real people will be able to use the approach when they have to know what other people are doing.
I have uploaded further comments in response to the authors reply.

Author Response

(The authors gave the same response as above.)

Reviewer 2 Report
Dear authors,
thank you for your responses to my questions and suggestions. You have mostly responded well, but in some cases the improvement has not gone to the manuscript. See my comments in the attached file: green text -> no changes requested; red text -> check and improve. In my opinion, this is minor revision.

Author Response

(The authors gave the same response as above.)
